# Investigation of the canine elbow joint innervation in 100 joints

**Jowita Jacewicz**[1☯]*, **Waldemar Sienkiewicz**[2☯], **Tomasz Burzykowski**[3],
**Beata Degórska**[1☯]*

1 Department of Small Animal Diseases and Clinic, Institute of Veterinary Medicine, SGGW in Warsaw, Warsaw, Poland, 2 Faculty of Veterinary Medicine, Department of Animal Anatomy, UWM in Olsztyn, Olsztyn, Poland, 3 Data Science Institute, Hasselt University, Hasselt, Belgium

☯ These authors contributed equally to this work.
* beata_degorska@sggw.edu.pl (BD); jowita_jacewicz@sggw.edu.pl (JJ)

**Data Availability Statement:** All relevant data are within the manuscript and its Supporting Information files.

**Funding:** The author(s) received no specific funding for this work.

## Abstract

The canine elbow joint is innervated by four nerves: the musculocutaneous, median, radial, and ulnar nerves. There is little data in the veterinary literature examining the course of the articular branches of those nerves. There is also no agreement as to their anatomical location in the joint capsule nor to their number. The lack of such data prompted us to undertake studies to analyze the course of these nerves and the number and location of their articular branches in both left and right joint capsules in 50 dogs (with a total of 100 joints). No within-individual differences in the number of branches or their location were found between left and right joints. There was between-joint variation in the number of articular branches of the median, musculocutaneous, and radial nerves. There was also variation in the location of the branches of the median and radial nerves. Some of the results are different from those previously reported in the literature. The obtained data allow to further detail anatomical descriptions of the dog elbow innervation. Additionally, they may constitute the basis for developing new surgical procedures, improving existing procedures, or allow a better preparation for possible differences during operations of this joint.

## Introduction

The canine elbow joint is innervated by the musculocutaneous, median, radial, and ulnar nerves [1,3]. Staszczyk et al. [1] reported that the elbow joint capsule features aspects of multiple, variable, asymmetric, and segmental innervation. The joint capsule is innervated by periosteal nerve, fibers originating from the muscles, and fibers that are branches of the main nerves innervating the joint.

The radial nerve receives most of its fibers from the C7–T1 nerve [4–7]. It leaves the axilla and in the brachial part it runs caudal and parallel to the brachial artery, then between the long and medial heads of the triceps. After detaching branches to the triceps reaches the lateral aspect of the humerus, heading to cranial aspect of the elbow [4–6]. The nerve is divided into a deep and superficial branch before leaving the brachium. The deep branch (*ramus profundus*) supplies all of the extensor muscles of the carpus—extensor carpi radialis (*m. extensor carpi*

**Competing interests:** NO authors have competing interests Enter: The authors have declared that no competing interests exist.

*radialis*), supinator muscle (*m. supinator*) and the digitis -common and lateral digital extensor (*m. extensor digitorum communis*, *m. extensor digitorum lateralis*), abductor digiti I longus (*m. abductor digiti I longus*) and extensor digiti I longus et digiti II. Then the deep branch supplies the antebrachiocarpal joint. Superficial branch of the radial nerve (*ramus superficialis*) supplies cutaneous area around and distal to the lateral epicondyle of the humerus, skin of the proximal one-third to two-thirds of the lateral surface of the antebrachium, skin on the cranial surface of the antebrachium, also innervate the dorsum of the manus [3,4,8]. An area of skin innervated by only one nerve is called an autonomous zone. Reduced or absent nociception to the skin in an autonomous zone implicates dysfunction of a specific nerve. For the skin of the manus, the autonomous zone for the radial nerve is the skin over the dorsal surface of digits III, IV [9,10].

The median nerve (C8–T1) after its origin from the brachial plexus, runs down the medial surface of the brachium. It combines with the musculocutaneous nerve to form a ramus communicans cum n. mediano around the axillary artery. At the cranial aspect of the elbow joint, the median nerve passes to the medial collateral ligament before dipping under the pronator teres (*m. pronator teres*) and flexor carpi radialis muscles (*m. flexor carpi radialis*). In the distal part of the antebrachium it divides into two or more branches, which descend through the carpal canal to innervate most of the structures on the palmar aspect of the distal thoracic limb. Median nerve gives muscular branches to the pronator teres, pronator quadratus (*m. pronator quadratus*), flexor carpi radialis, flexor digitorum superficialis *(m. flexor digitorum superficialis)*, radial head of the flexor digitorum profundus (*m. flexor digitorum profundus caput radiale)*, deep part of the humeral head of the flexor digitorum profundus (*m. flexor digitorum profundus caput humerale*) [3,4,8]. The median nerve innervates the skin of the palmar surface of the metacarpus and fingers [3].

The ulnar nerve (C8–T1) [6,7] runs distally on the medial aspect of the antebrachium caudal to the median nerve and brachial artery. In the distal half of the brachium it seeks a more caudal course over the medial epicondyle of the humerus, running under the ulnar head of the ulnar flexor muscle to the ulnar groove on the caudal aspect of the antebrachium [5,6]. Ulnar nerve gives muscular branches to the flexor carpi ulnaris (*m. flexor carpi ulnaris*), flexor digitorum profundus, flexor digitorum superficialis, interosseus (*mm. interossei*) [3,8,11]. It supplies to the skin across the proximal portion of the antebrachium from the medial to the caudolateral aspects, branches to the skin of the distal part of antebrachium,proximal two-thirds of the skin of the caudolateral aspect of the antebrachium [3]. The main trunk of the ulnar nerve divides into dorsal and palmar branches in the distal half of the antebrachium. The dorsal branch innervates the skin on the lateropalmar aspect of the manus. The palmar branch crosses the carpus with the flexor tendons and median nerve to supply the palmar aspect of the manus [6]. The autonomous zone for the ulnar nerve is the skin over the lateral aspect of digit V [9,10].

The musculocutaneous nerve originates from C 7 [12]. The nerve arises from the brachial plexus. It runs parallel to the median nerve and detaches a communicating branch to it. In the proximal part of the humerus, musculocutaneous nerve gives muscular branches to the coracobrachialis (*m. coracobrachialis*) and biceps brachii (*m. biceps brachii*). In the distal third of the upper brachium, musculocutaneous nerve innervates the brachial muscle (*m. brachialis*) [3–6] and the skin on the medial aspect of the antebrachium [3,5,6]. The autonomous zone for the musculocutaneous nerve is the skin over the 2 cm distal to the medial epicondyle of the humerus [9,10].

There are few publications in the veterinary literature analyzing the course of the aforementioned nerves towards the capsule of the canine elbow joint. Of those available, several contain information about the articular branches of those nerves [1,2,13]. There is no consensus as to

the anatomical location of the branches of individual nerves within the joint capsule nor their number. This article aims to expand the current knowledge regarding the innervation of the dog's elbow.

## Materials and methods

The aim of the study was to examine the course of all four nerves participating in the innervation of the canine elbow joint, determine their location within the joint capsule, and investigate the number of articular branches of individual nerves. Due to differences in breed and size of individuals, the diameter of the articular branches was not measured. The research was performed on isolated thoracic limbs of dogs that had previously been euthanized in veterinary clinics for reasons that excluded elbow joint diseases. Pursuant to Article 1(2) of the Act on the Protection of Animals Used for Scientific or Educational Purposes, veterinary services and clinical veterinary tests are not experimental procedures and do not require the consent of the Ethics Committee. The examinations were carried out on isolated right and left thoracic limbs of 50 dogs (23 males and 27 females, from six to fifteen years of age, with 14 (28%) dogs aged 10 or younger) of 19 different breeds (see Table 1). The course of the four nerves innervating the dog elbow joint was analyzed by using a dissecting loupe with 2.5x magnification (Seliga Optical) and basic surgical instruments (tissue forceps, anatomical tweezers, blades, Metzenbaum scissors and a Gelpi retractor). Tissue preparation began on the medial side of the limb. After cutting the skin and fascia above the humeral head, the ulnar nerve was identified. It was dissected from the surrounding tissue dorsally and ventrally to the elbow joint capsule along the medial head of the triceps brachii muscle. At the level of the elbow joint, the nerve ran behind the condylus of the humerus, under the flexor digitorum superficialis muscle. Dissection of the median nerve began from the axillary region ventrally to the elbow joint capsule. In the shoulder area, the median nerve ran between the ulnar nerve and the musculocutaneous nerve. During the preparation, just above the elbow joint, ramus communicans cum n. mediano was found–a connection between the musculocutaneous and median nerves. The nerve was dissected into the joint capsule of the elbow in the area of the pronator teres muscle and the flexor carpi radialis muscle. The course of the musculocutaneous nerve was examined from the axillary region towards the ventral. The connection between the median nerve and the musculocutaneous nerve mentioned earlier facilitated the identification of this nerve. At the level of this connection, the nerve was isolated from the superficial brachial artery and surrounding tissues towards the elbow joint capsule. This nerve also gave off branches to the brachialis muscle. The radial nerve, in turn was searched for on the lateral side of the brachium. The nerve was dissected from the axillary region and then along the brachialis muscle towards

**Table 1. The breeds of dogs included in the examination, classified by size (based on dog's actual weight).**

| Small, <10 kg (number of dogs) | Medium, 10–26 kg (number of dogs) | Large, 26+ kg (number of dogs) |
|---|---|---|
| Chihuahua (1) | Beagle (1) | American Staffordshire Terrier (1) |
| Maltese (2) | Cocker Spaniel (1) | Bernese Mountain Dog (1) |
| Pug (1) | Dachshund (1) | Dalmatian (1) |
| Shih Tzu (1) | French Bulldog (3) | German Shepherd (3) |
| Standard Poodle (1) | Mix (19) | Irish Setter (1) |
| West Highland White Terrier (1) | | Labrador Retriever (1) |
| Yorkshire Terrier (1) | | Pointer (1) |
| Mix (4) | | Mix (4) |

**Table 2. Number of articular branches of the nerves innervating the dog's right and left elbow joint for the 50 examined dogs.**

| Left joint branches | Ulnar nerve | | | Musculocutaneous nerve | | | Median Nerve | | | Radial nerve | | |
|---|---|---|---|---|---|---|---|---|---|---|---|---|
| | Right joint branches | | | Right joint branches | | | Right joint branches | | | Right joint branches | | |
| | 1 | 2 | 3 | 1 | 2 | 3 | 1 | 2 | 3 | 1 | 2 | 3 |
| 1 | 50 | 0 | 0 | 10 | 0 | 0 | 20 | 0 | 0 | 33 | 0 | 0 |
| 2 | 0 | 0 | 0 | 0 | 39 | 0 | 0 | 19 | 0 | 0 | 10 | 0 |
| 3 | 0 | 0 | 0 | 0 | 0 | 1 | 0 | 0 | 11 | 0 | 0 | 7 |

the elbow joint. At the level of the joint, the nerve reached the cranial side, continuing along the extensor carpi radialis muscle.

95% confidence interval (95% CI) for the proportion of joints containing a particular number of branches or for the proportion of branches in a particular location was estimated by assuming the multinomial distribution for matched pairs of categorical data [14] while taking into account the association between the outcomes for the left and right joint for each dog. The distribution of dogs in relation to the number of branches per joint were compared by using the exact version of Pearson's chi-squared test [14].

## Results

### Ulnar nerve

During its course, the ulnar nerve was found to encircle the medial epicondyle of the humerus caudally. In all (100% of) the examined joints, the nerve sent one branch (see Table 2), always to the joint capsule of the elbow in its caudal medial part (Figs 1–3) after caudal retraction of the flexor carpi ulnaris muscle, between the olecranon and the medial epicondyle of the humerus, most often at ½ of the length just below the medial epicondyle.

### Median nerve

In both joints of 48 of the 50 examined dogs, i.e., in 96% (95% CI) [90.6%, 100%]) of the 100 examined joints, the median nerve presented articular branches between the pronator teres muscle and the flexor carpi radialis muscle, above the medial epicondyle, at its ½ length, reaching the medial quadrant of the joint capsule (Figs 4 and 5). In only two dogs, i.e., in 4% (95% CI [0%, 9.4%]) of the examined joints, these branches reached the medio-cranial segment of the joint capsule. Three patterns showing different number of articular branches of this nerve were observed, with the same number of branches in the left and right joint for any particular dog (see Table 2). Most often there was one articular branch, as observed in 40 (40%, 95% CI [26.4%, 53.6%]) of the examined joints. In 38 (38%, 95% CI [24.5%, 51.4%]) joints there were two branches, and three in 22 (22%, 95% CI [10.5%, 33.5%]) joints (Fig 6).

The distribution of dogs in relation to the number of branches per joint was statistically significantly ($p = 0.018$) different for the two sexes (see Table 3). There was no statistically significant difference (see S1 and S2 Tables) by breed/size ($p = 0.43$) nor by age (younger or older than 11 years, $p = 0.54$).

### Musculocutaneous nerve

Along the its course, the nerve presented articular branches on the medio-cranial side of the elbow joint capsule under the brachialis muscle (Figs 7 and 8) in all (100% of) the examined joints. Three patterns showing different number of articular branches of this nerve were

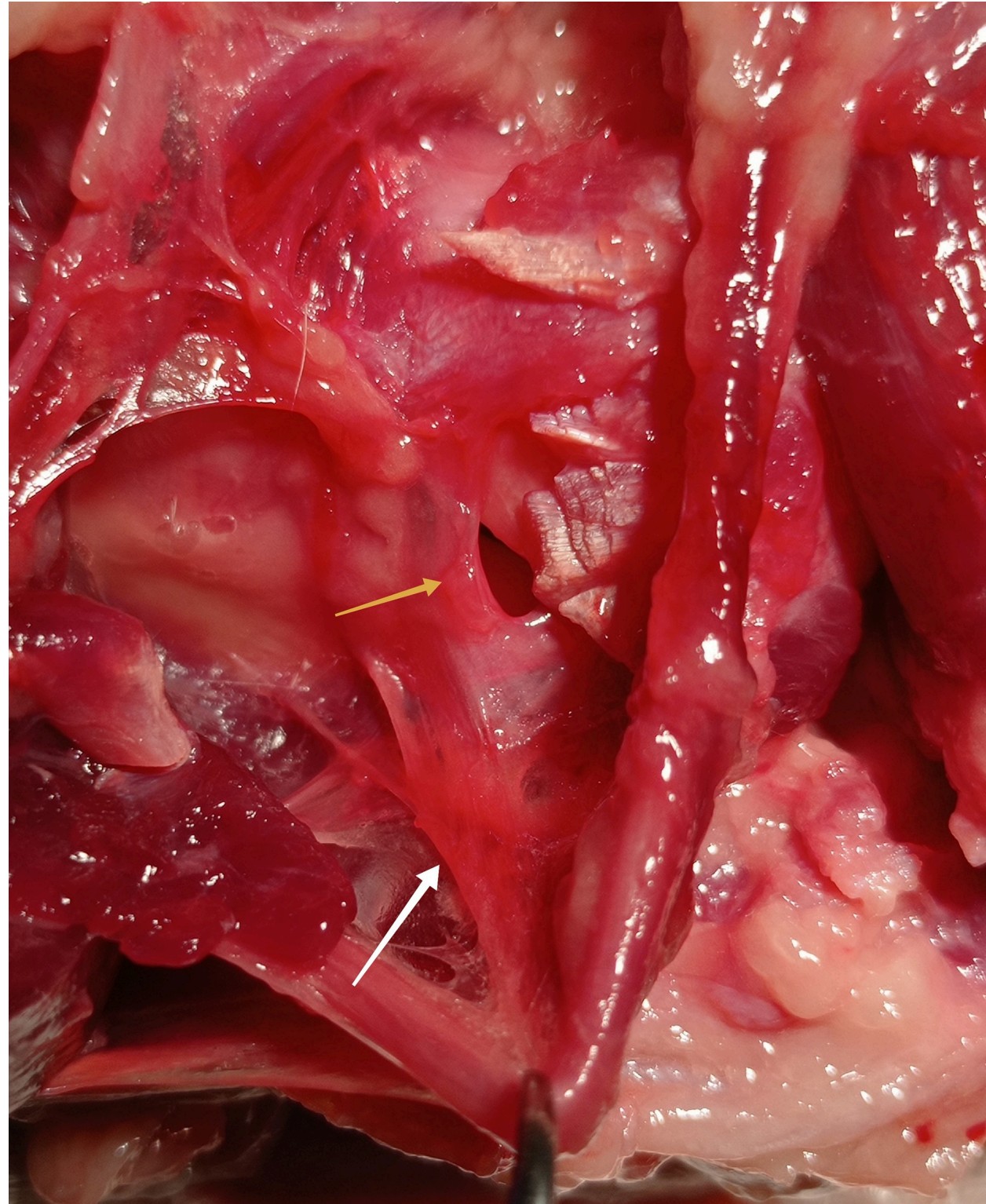

**Fig 1. Branch of the ulnar nerve (white arrow) to the elbow capsule (yellow arrow) of the dog.** Photo taken by the author.

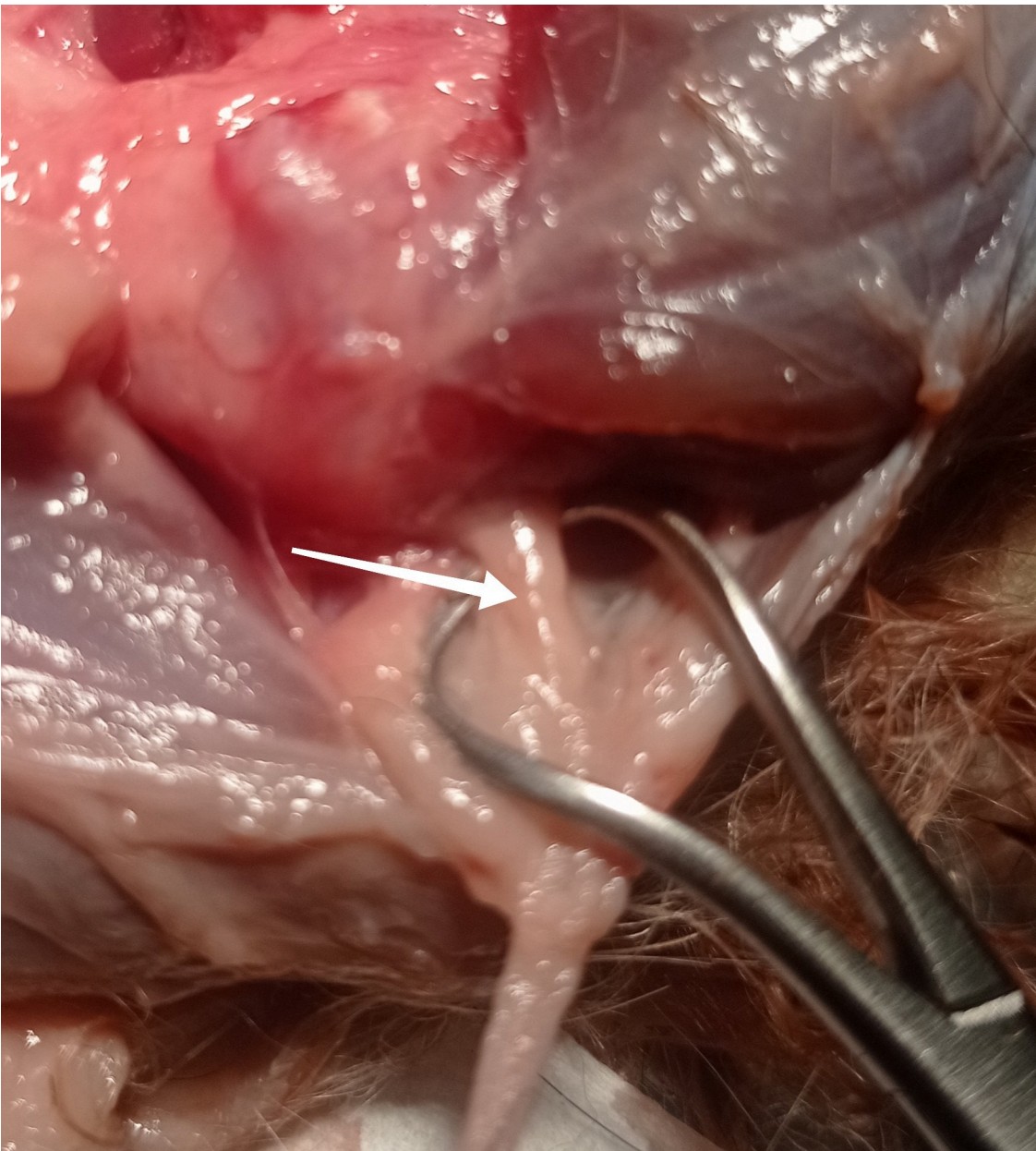

**Fig 2. Articular branch (white arrow) of the ulnar nerve into caudomedial quadrant of the elbow joint.** Photo taken by the author.

observed, with the same number of branches in the left and right joint for any particular dog (see Table 2). In 20 (20%, 95% CI [8.9%, 31.1%]) of the examined joints there was one direct branch to the elbow joint capsule. In 78 (78%, 95% CI [66.5%, 89.5%]) of the cases the nerve was divided into two branches. Three branches of the musculocutaneous nerve were observed in one individual dog (Fig 9), i.e., in two joints (2%, 95% CI [0%, 5.9%]).

There was no statistically significant difference ($p = 0.72$) in the distribution of dogs in relation to the number of branches per joint for the two sexes (see Table 3). Similarly, there was no statistically significant difference (see S3 and S4 Tables) by breed/size ($p = 0.17$) nor by age ($p = 0.58$).

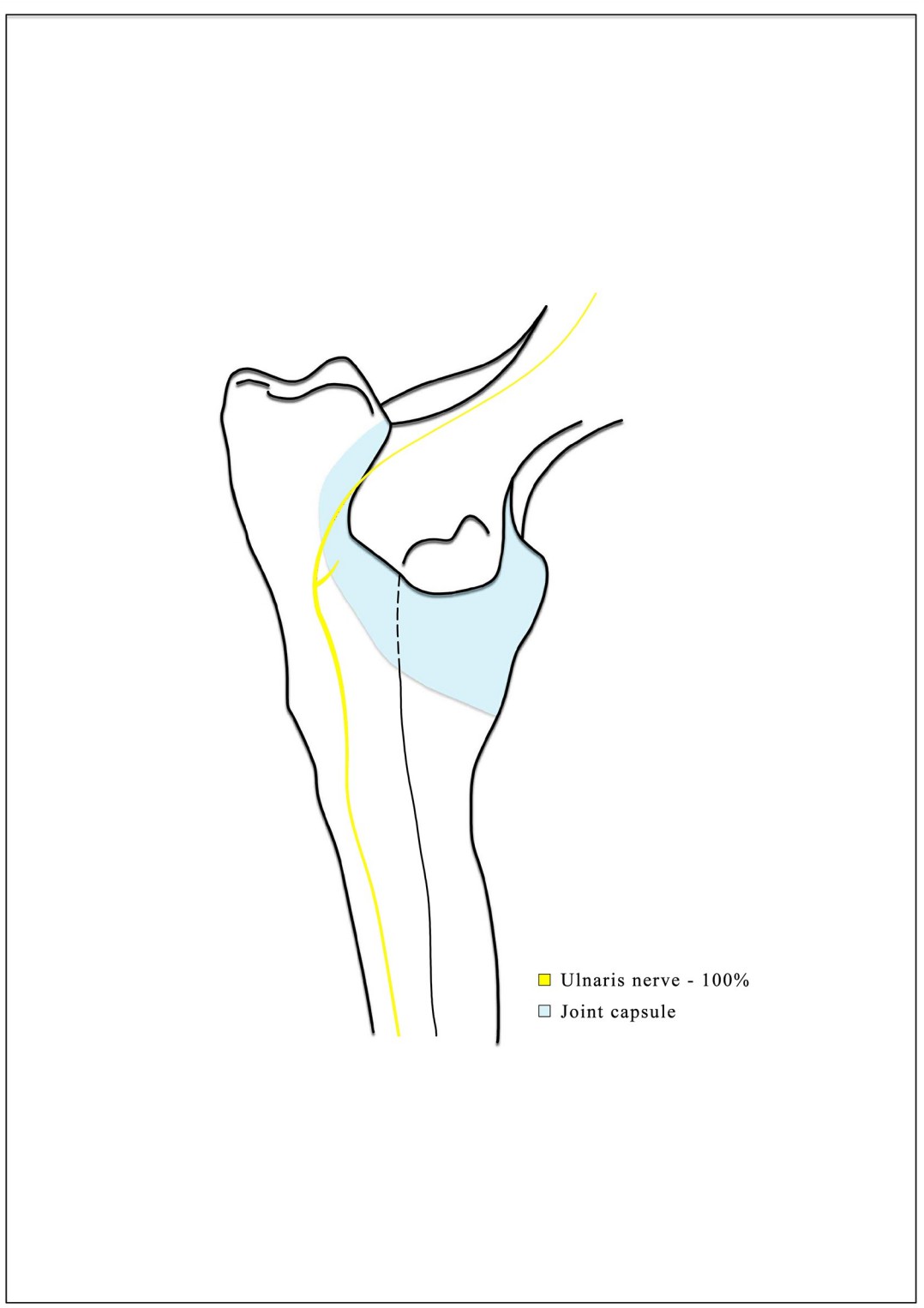

**Fig 3. Diagram showing the articular branch of the ulnar nerve.** The graphic prepared by the author.

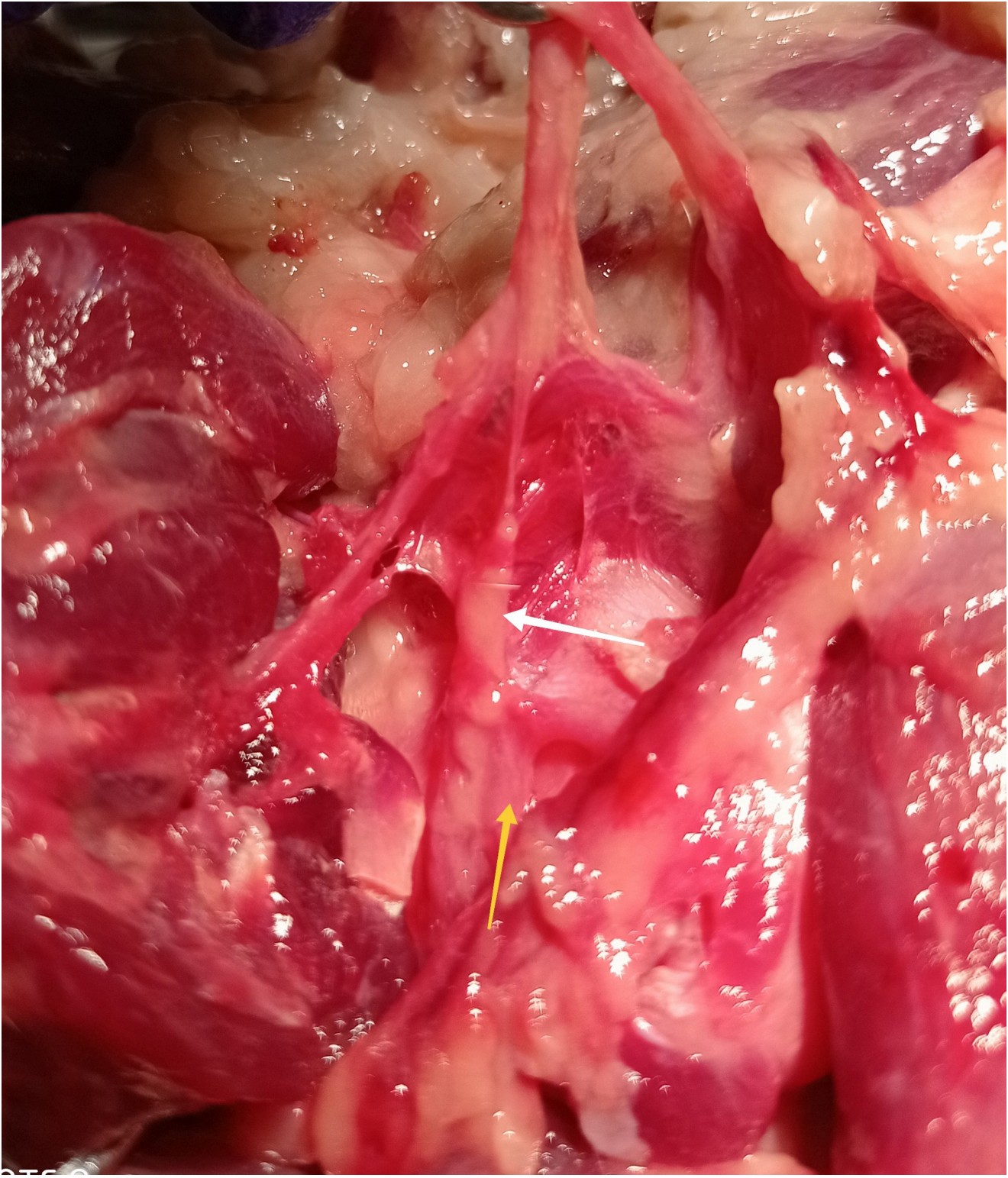

**Fig 4. Branch of the median nerve to the elbow joint capsule of the dog. (white arrow—articular branch of the nerve; yellow arrow—joint capsule).** Photo taken by the author.

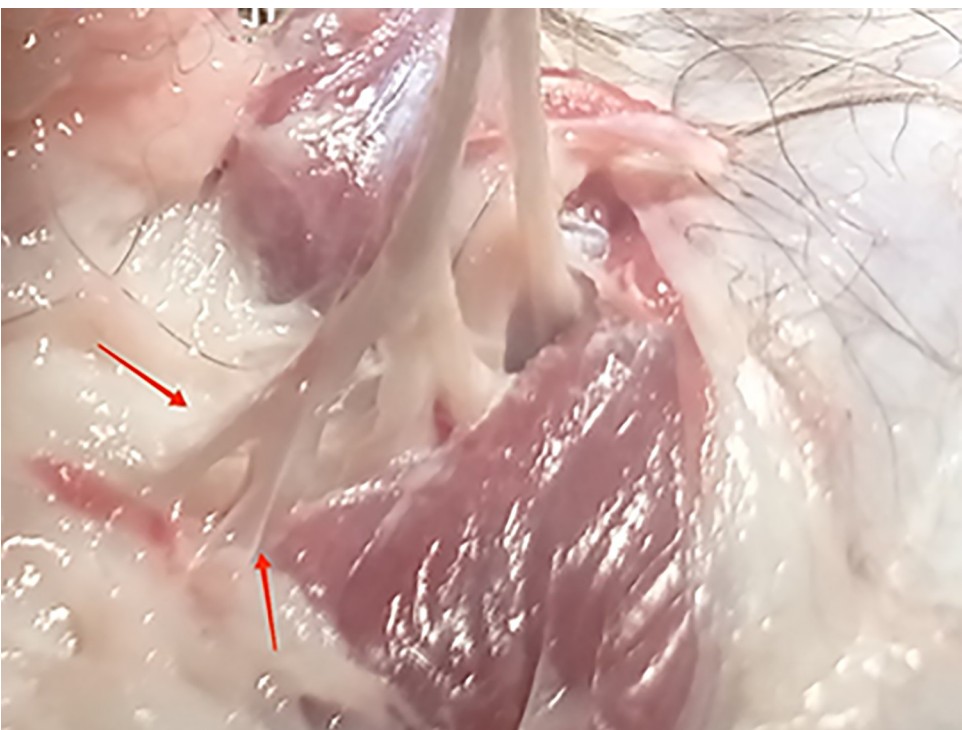

**Fig 5. Articular branches (red arrows) of the median nerve.** Photo taken by the author.

## Radial nerve

The radial nerve transmitted articular branches to the cranial quadrant of the capsule, between the extensor carpi radialis and the common digital extensor muscles *(musculus extensor digitorum communis)* (Fig 10), in both joints of 40 dogs, i.e., in 80% (95% CI [68.9%, 91.1%]) of the examined joints. In the remaining 10 dogs, i.e., 20% (95% CI [8.9%, 31.1%]) of the joints,

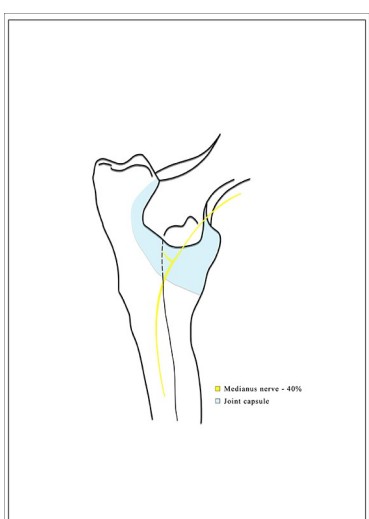 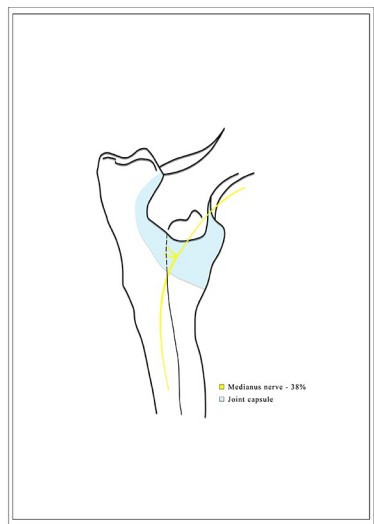 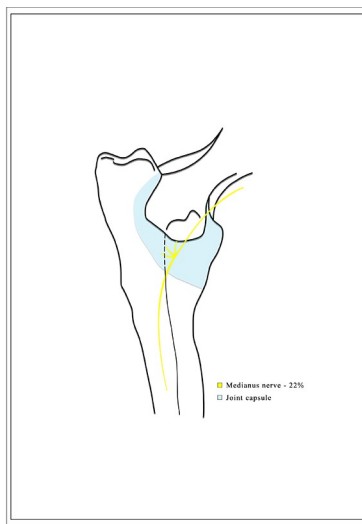

**Fig 6. Diagram showing the articular branches of the median nerve.** The graphic prepared by the author.

**Table 3. Number (percentage) of dogs with different number of articular branches per joint by sex.** (in brackets: p-values for the exact version of Pearson's chi-squared test comparing the distribution for the sexes).

| Sex | Musculocutaneous nerve [p = 0.72] | | | Median nerve [p = 0.018] | | | Radial nerve [p = 0.001] | | |
|---|---|---|---|---|---|---|---|---|---|
| | Number of branches | | | Number of branches | | | Number of branches | | |
| | 1 | 2 | 3 | 1 | 2 | 3 | 1 | 2 | 3 |
| Female | 6 (22.2%) | 21 (77.8%) | 0 | 7 (25.9%) | 15 (55.6%) | 5 (18,5%) | 12 (44.4%) | 9 (33.3%) | 6 (22.2%) |
| Male | 4 (17.4%) | 18 (78.3%) | 1 (4.3%) | 13 (56.5%0 | 4 (17.4%) | 6 (26.1%) | 21 (91.3%) | 1 (4.3%) | 1 (4.3%) |

the branches reached the craniolateral area of the joint capsule. During preparation, three patterns were observed showing the articular branches of the radial nerve reaching the elbow joint capsule, with the same number of branches in the left and right joint for any particular dog (see Table 2). In 66 (66%, 95% CI [52.9%, 79.1%]) of the examined joints, there was one main branch of the radial nerve just above the humeroradial joint; in 20 (20%, 95% CI [8.9%, 31.1%]) joints there were two branches; and in 14 (14%, 95% CI [4.4%, 23.6%]) there were three articular branches of the radial nerve (Fig 11).

The number of dogs in relation to the number of branches per joint was statistically significantly ($p = 0.001$) different for the two sexes (see Table 3). There was no statistically significant difference (see S5 and S6 Tables) by breed/size ($p = 0.12$) nor by age ($p = 0.91$).

## Discussion

The location of the articular branches within the joint capsule and their number were described by Staszyk et al. [1] and ten years later by Yilmaz and Bahadir [2]. The former evaluated 19 thoracic limbs of 11 dogs, the latter provided findings for 10 thoracic limbs of five Turkish Shepherd dogs. Table 4 presents the summary of the results reported in Yilmaz and Bahadir's study [2]. It is not possible to present such a summary for the results reported in Staszyk's study [1] due to the limited information provided in that publication.

In older publications, the location of the articular branches of the ulnar nerve was described in the caudal [15] or caudomedial [16] part of the joint capsule between the trochlea humeri and the olecranon. According to more recent data, the ulnar nerve in dogs sends articular branches to the medial area of the elbow joint capsule [1,2,11], located in its central part, between the trochlea of the humerus and the olecranon and between the ulnar head of the flexor carpi ulnaris muscle and the superficial digital flexor. In results reported in Yilmaz and Bahadir's study [2] the nerve usually give off one (80% of the examined limbs) or two (20% of the examined limbs) articular branches (see Tables 4 and 5). In our study, the ulnar nerve was found, as reported in Staszyk's study [1], to always give off one articular branch (see Tables 2 and 5) in a similar location (see Table 6) to that specified in [1] and [2].

The median nerve runs along the medial surface of the brachium towards the condylus humeri. This nerve reaches the pronator teres muscle and continues caudally to the group of antebrachium flexor muscles. According to results reported in Staszyk, Yilmaz and Bahadir, Hermanson's study[1–3], the median nerve gives rise to one articular branch (see Tables 4 and 5). According to Hermanson et al. [3], it is located in the medial part of the elbow capsule. However, Staszyk et al. [1] and Yilmaz and Bahadir [2] describe the intra-articular branch reaching the cranial segment of the joint capsule. Additionally, Yilmaz and Bahadir [2] found the articular branches of the nerve reach the craniomedial part of the joint capsule in one (10%) of the 10 examined joints (see Table 6). In our study, the nerve was found in the medial

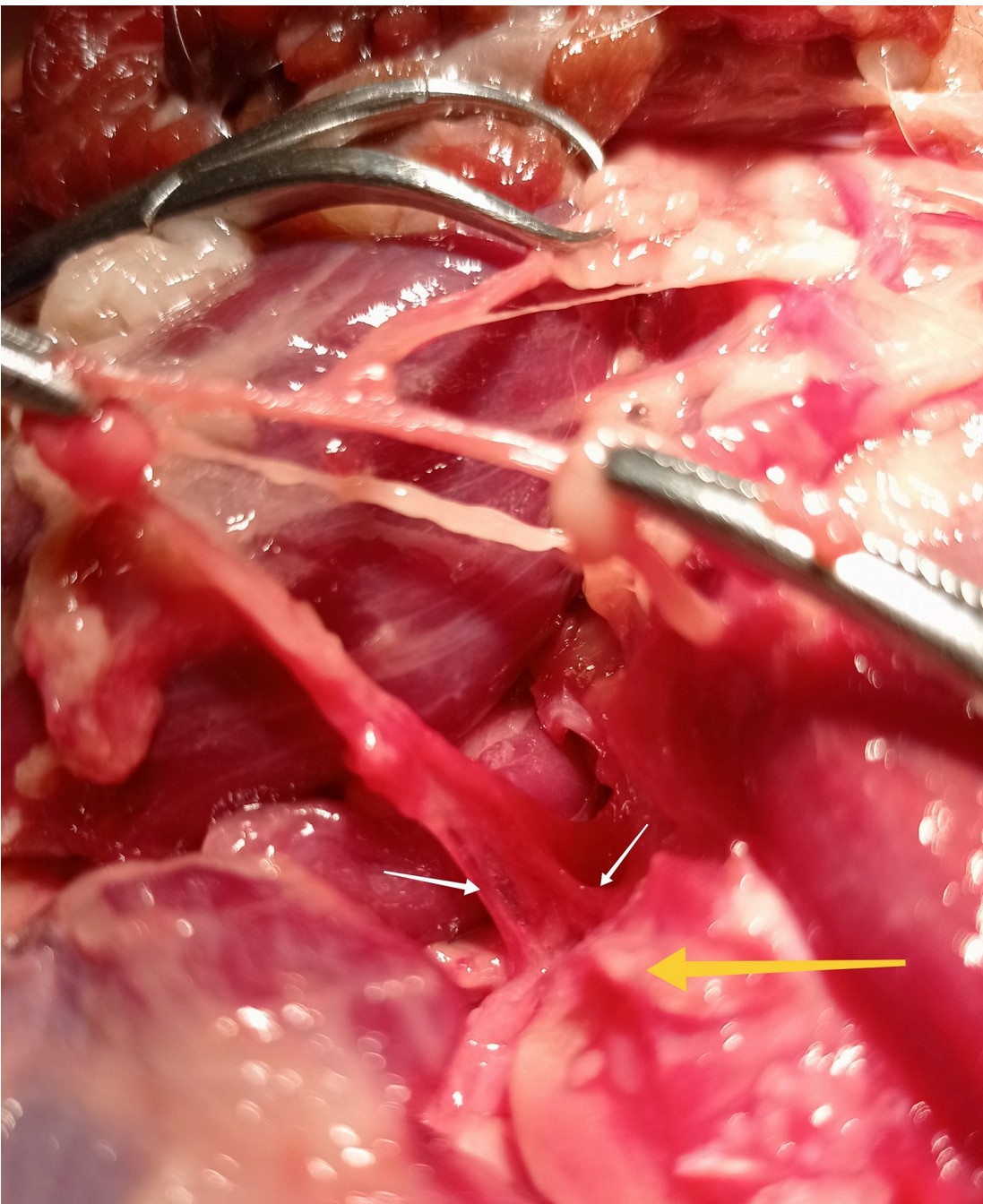

**Fig 7. Articular branches of musculocutaneous nerve (white arrow—articular branch of the nerve; yellow arrow—joint capsule).** Photo taken by the author.

section of the joint capsule in 96% of the joints (see Table 6). There was variability in the number of branches per joint: one branch was found in 40%, two branches in 38%, and three branches in 22% of the examined joints (see Tables 2 and 5).

The musculocutaneous nerve, upon reaching the distal, medial part of the biceps brachii muscle, divided into the distal muscular branch of the nerve (called the nerve to brachialis muscle) and a small medial cutaneous nerve of the thoracic limb. There is no consensus in the

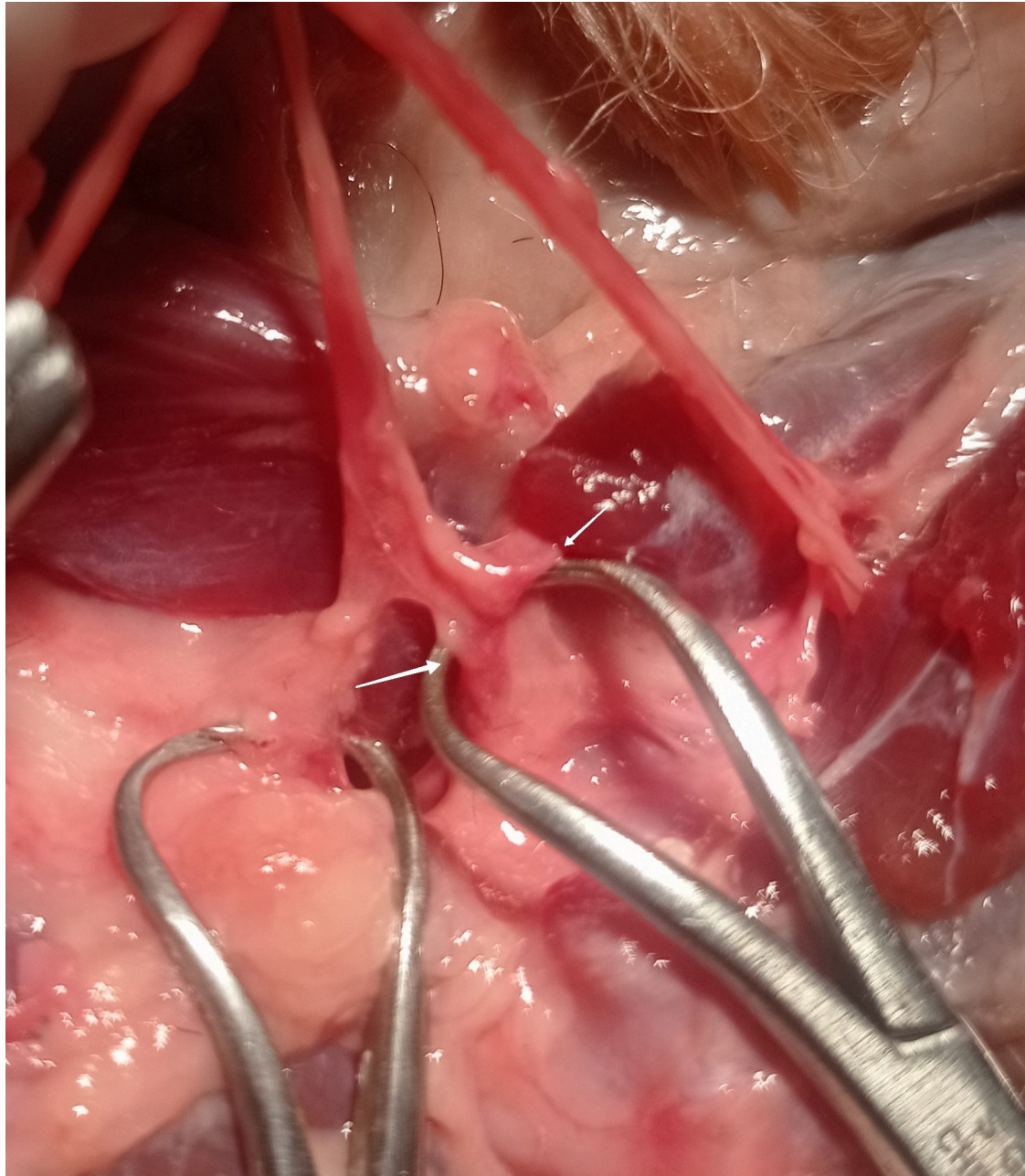

**Fig 8. Articular branches (white arrow) of musculocutaneous nerve to the mediocranial quadrant of the elbow joint capsule.**
Photo taken by the author.

literature regarding the articular branches of the musculocutaneous nerve. As an example of such differences, we find that the 3rd edition of Miller's Atlas of Anatomy reports that the articular branch comes from the distal muscular branch of the musculocutaneous nerve [17]. But the latest, 5th edition, of this atlas states that the branch comes from the medial cutaneous nerve of the antebrachium [3]. There is also information about one articular branch of the nerve in the craniolateral part of the elbow joint capsule. According to Staszyk et al.[1] and Yilmaz and Bahadir [2], the articular branches usually reach the cranial area of the elbow joint

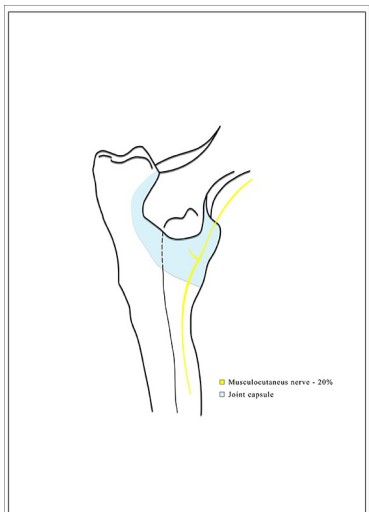
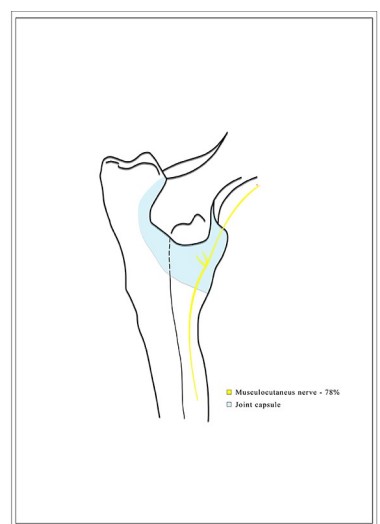
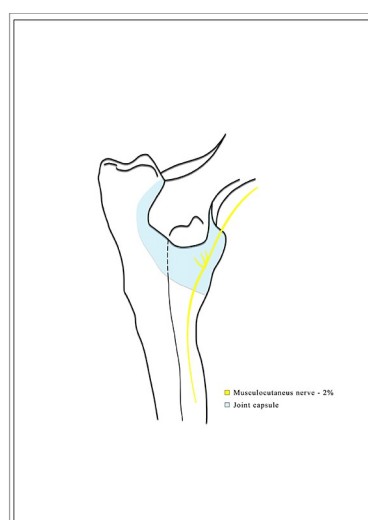

**Fig 9. Diagram showing the articular branches of the musculocutaneous nerve.** The graphic prepared by the author.

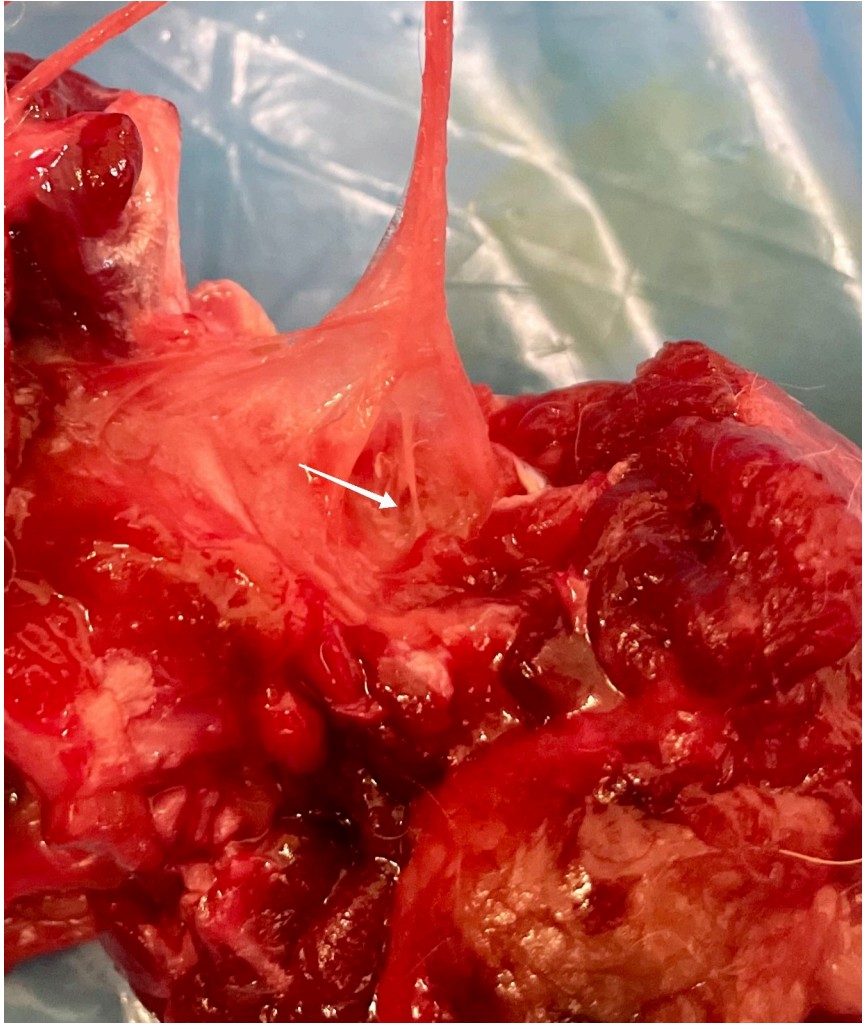

**Fig 10. Articular branch (white arrow) of radial nerve.** Photo taken by the author.

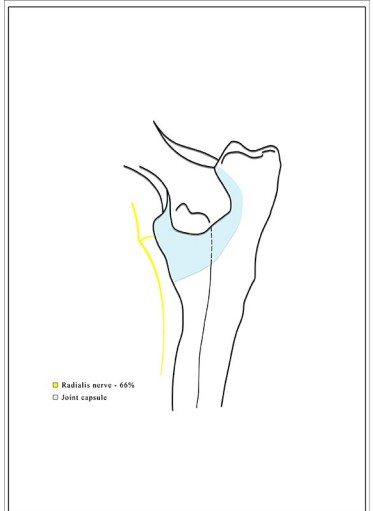 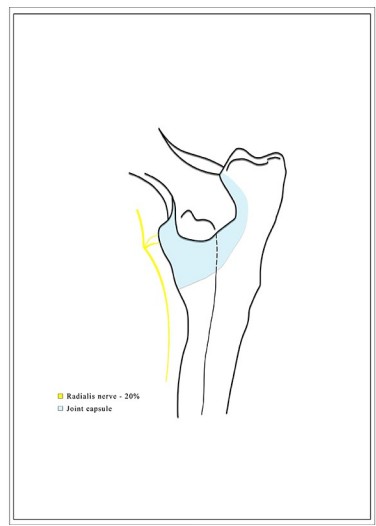 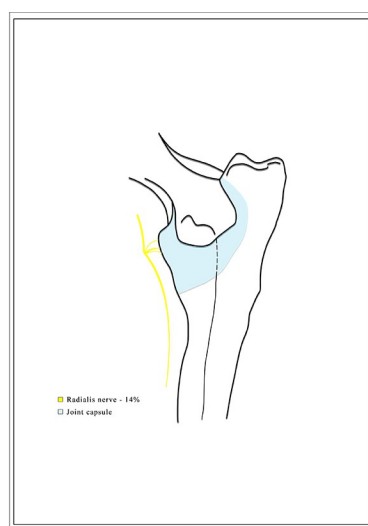

**Fig 11. Diagram showing the articular branches of the radial nerve.** The graphic prepared by the author.

capsule (see Table 6), slightly lateral and proximal to the biceps tendon. In the examined preparations, we observed all (100% of) articular branches of this nerve in the craniomedial part of the elbow joint capsule (see Table 6). Staszyk et al. [1] observed that direct articular branches of this nerve were rare (in 3 out of 19 isolated limbs). In most cases (84%), the branches of the musculocutaneous nerve formed two or three secondary branches (see Table 5). In turn, Yilmaz and Bahadir [2] found one direct branch to the joint capsule occurring in half of the examined limbs, while two secondary branches were isolated in the remaining half of the joints (see Tables 4 and 5). In our study, the musculocutaneous nerve gave one direct branch to the capsule of the elbow joint in 20 (20%) of the 100 examined joints; in 78 (78%) of the joints the nerve was divided into two branches; and it was divided in three branches in only two (2%) joints (see Tables 2 and 5).

After reaching the lateral head of the triceps muscle *(m. triceps brachii, caput laterale)*, the radial nerve branches into deep and superficial branches. In available sources, only one, the deep branch of the radial nerve, gave an articular branch. Miller's Anatomical Atlas [17] shows that when the deep branch crosses the flexor surface of the elbow joint, it sends the articular branch to the craniolateral surface of the joint. Yilmaz and Bahadir [2] found that the radial nerve usually (in 90% of the examined 10 joints) supplied the craniolateral part of the joint capsule. Only in 10% of cases (one joint) the nerve gave off one articular branch ultimately innervating the cranial area of the joint capsule [2]. In our study, the radial nerve supplied the cranial area of the capsule of the elbow joint in 80% of the examined joints, and in the remaining 20% it supplied the craniolateral part (see Table 6). In results reported in Staszyk et al. [1],

**Table 4. Number of articular branches of the nerves innervating the dog's right and left elbow joint for five Turkish Shepherd dogs examined in [2].**

| Left joint branches | Ulnar | | | Musculocutaneous | | | Median | | | Radial | | |
|---|---|---|---|---|---|---|---|---|---|---|---|---|
| | Right joint branches | | | Right joint branches | | | Right joint branches | | | Right joint branches | | |
| | 1 | 2 | 3 | 1 | 2 | 3 | 1 | 2 | 3 | 1 | 2 | 3 |
| 1 | 3 | 0 | 0 | 2 | 0 | 0 | 5 | 0 | 0 | 4 | 0 | 0 |
| 2 | 2 | 0 | 0 | 1 | 2 | 0 | 0 | 0 | 0 | 0 | 1 | 0 |
| 3 | 0 | 0 | 0 | 0 | 0 | 0 | 0 | 0 | 0 | 0 | 0 | 0 |

**Table 5. Percentages (with 95% confidence intervals) of joints with a particular number of articular branches of the nerves reported in [1] and [2] and observed in the current study.**

| Nerve | Branches | Ref [1]* | Ref [2]** | Current study |
|---|---|---|---|---|
| Ulnar | 1 | 100% | 80% [58.5%, 100%] | 100% |
| | 2 | | 20% [0%, 41.5%] | |
| | 3 | | | |
| Median | 1 | 100% | 100% | 40% [26.4%, 53.6%] |
| | 2 | | | 38% [24.5%, 51.4%] |
| | 3 | | | 22% [10.5%, 33.5%] |
| Musculocutaneous | 1 | 16% | 50% [10.8%, 89.2%] | 20% [8.9%, 31.1%] |
| | 2 | 84% | 50% [10.8%, 89.2%] | 78% [66.5%, 89.5%] |
| | 3 | | | 2% [0%, 5.9%] |
| Radial | 1 | 60% | 80% [44.9%, 100%] | 66% [52.9%, 79.1%] |
| | 2 | 40% | 20% [0%, 55.1%] | 20% [8.9%, 31.1%] |
| | 3 | | | 14% [4.4%, 23.6%] |

* For [1], confidence intervals cannot be provided because of the lack of the necessary data

** For [2], confidence intervals were computed based on the data in Table 4.

Yilmaz and Bahadir's study [2], the branches opened into the capsule of the dog's elbow joint proximally or distally to the annular ligament.

The reported number of articular branches of the nerve varies in the literature. According to Yilmaz and Bahadir [2], in 80% (8 of the 10 examined elbow joints) of cases, the nerve gives off one articular branch, and in 20% two articular branches (see Tables 4 and 5). In macroscopic studies conducted by Staszyk et al. [1], one articular branch was observed in 60% and two articular branches in 40% of joints (see Table 5). In our study (see Tables 2 and 5), we found that the nerve gave off mainly (66% of the 100 examined joints) one branch, with two branches observed in 20 (20%) of the examined joints, and three in 14 (14%) of the joints. When the nerve gave off more than one branch, one of them was stronger.

The results of the macroscopic examinations confirm that the dog's elbow joint is innervated by the ulnar, median, radial, and musculocutaneous nerves. We did not observe differences between the left and right joints of a given individual with respect to the number (see

**Table 6. Percentages (with 95% confidence intervals) of joints with branches reaching a particular location reported in [1] and [2] and observed in the current study.**

| Nerve | Location | Ref [1]* | Ref [2]** | Current study |
|---|---|---|---|---|
| Ulnar | medial | 100% | 80% [58.5%, 100%] | 100% |
| | caudo-medial | | 20% [0%, 41.5%] | |
| Median | cranio-medial | | 10% [0%, 27.5%] | 4% [0%, 9.4%] |
| | medial | | | 96% [90.6%, 100%] |
| | cranial | 100% | 90% [72.5%, 100%] | |
| Musculocutaneous | cranial | 100% | 100% | |
| | cranio-medial | | | 100% |
| Radial | cranial | cranial | 10% [0%, 27.5%] | 80% [68.9%, 91.1%] |
| | cranio-lateral | | 90% [72.5%, 100%] | 20% [8.9%, 31.1%] |

* For [1], confidence intervals cannot be provided because of the lack of the necessary data

** For [2], confidence intervals were computed based on the data provided in [1].

Table 2) and location of the articular branches. This is in contrast to the results reported in Yilmaz and Bahadir's study [2], where some within-individual differences were observed (see Table 4).

Taking into account the precision of the estimated percentages, the distribution of the locations of the articular branches of the ulnar nerve, observed in our study, was comparable (see Table 6) to those reported in Staszyk et al. [1],Yilmaz and Bahadir's study [2]. For the median nerve, we found the articular branches of this nerve mainly (96%) in the medial quadrant of the joint capsule. This is in line with what was described in Hermanson [3], but in contrast to Staszyk, Yilmaz and Bahadir's study [1,2], where the authors indicated mainly cranial location (see Table 6). For the musculocutaneous nerve, we observed 100% of branches in craniomedial location, while Yilmaz and Bahadir [2] reported 100% branches in cranial location. Finally, in our study, branches of the radial nerve were mainly (80%) found in craniolateral location, while Yilmaz and Bahadir [2] reported that location in only 10% of the cases (see Table 6).

Regarding the number of articular branches per joint, the results obtained in our study (see Table 5) for the ulnar and musculocutaneous nerve are comparable, when taking into account the precision of the estimated percentages, to those published in the literature. For the radial nerve, our results suggest, unlike in Staszyk, Yilmaz and Bahadir [1,2], the possibility of a nonzero (4.4% to 23.6%; see Table 5) fraction of joints with three branches. The most substantial difference, as compared to Staszyk, Yilmaz and Bahadir [1,2], was noted for the median nerve, for which our study indicates that the fraction of joints with one branch may be between 26.4% and 53.6% rather than 100% (see Table 5); moreover, there may be 24.5% to 51.4% of joints with two branches and 10.5% to 33.5% joints with three branches.

The source of these differences is unclear; it may be, for instance, the different mix of breeds included in our study, as compared to Staszyk,Yilmaz and Bahadir [1,2].

Interestingly, we found a statistically significant difference in the number of dogs in relation to the number of branches per joint for sex for the median ($p = 0.018$) and radial ($p = 0.001$) nerves. In both cases, there was a substantially larger proportion of male dogs with only one branch per nerve as compared to females. No such differences were reported previously in the literature. To rule out the play of chance, the finding should be validated in an independent study. We did not find any statistically significant difference in the distribution of dogs in relation to the number of branches per joint for breed/size or age.

It is worth noting that our research was carried out on the right and left elbow joints of 50 individuals, i.e., using 100 joints in total. The studies by Staszyk et al. [1] and Yilmaz and Bahadir [2] were conducted on much smaller groups including,11 dogs (19 joints in total) and 5 dogs (10 joints in total), respectively. As a result, the level of precision achieved in our study is higher as compared to the two other studies, as can be seen from the narrower confidence intervals provided in Tables 5 and 6 for our estimates. We believe that the reported, more precise results expand the knowledge about the innervation of the dog's elbow joint, which may be useful in further clinical research, especially in the field of surgical procedures.

## Supporting information

**S1 Table. Number (percentage) of dogs with different number of articular branches per joint for the median nerve by breed/size.**
(PDF)

**S2 Table. Number (percentage) of dogs with different number of articular branches per joint for the median nerve by age.**
(PDF)

**S3 Table. Number (percentage) of dogs with different number of articular branches per joint for the musculocutaneous nerve by breed/size.**
(PDF)

**S4 Table. Number (percentage) of dogs with different number of articular branches per joint for the musculocutaneous nerve by age.**
(PDF)

**S5 Table. Number (percentage) of dogs with different number of articular branches per joint for the radial nerve by breed/size.**
(PDF)

**S6 Table. Number (percentage) of dogs with different number of articular branches per joint for the radial nerve by age.**
(PDF)

## Acknowledgments

The authors would like to thank all who have contributed to the study conducted.

## Author Contributions

**Conceptualization:** Jowita Jacewicz, Waldemar Sienkiewicz, Beata Degórska.

**Data curation:** Jowita Jacewicz.

**Formal analysis:** Jowita Jacewicz, Waldemar Sienkiewicz, Tomasz Burzykowski, Beata Degórska.

**Funding acquisition:** Jowita Jacewicz, Beata Degórska.

**Investigation:** Jowita Jacewicz.

**Methodology:** Jowita Jacewicz, Beata Degórska.

**Project administration:** Jowita Jacewicz, Waldemar Sienkiewicz, Beata Degórska.

**Resources:** Jowita Jacewicz.

**Software:** Tomasz Burzykowski.

**Supervision:** Waldemar Sienkiewicz, Tomasz Burzykowski, Beata Degórska.

**Validation:** Waldemar Sienkiewicz.

**Visualization:** Jowita Jacewicz, Waldemar Sienkiewicz, Beata Degórska.

**Writing – original draft:** Jowita Jacewicz.

**Writing – review & editing:** Jowita Jacewicz, Tomasz Burzykowski, Beata Degórska.

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
