## [Decision Letter · Decision Letter 0]

12 Aug 2024

PONE-D-24-21054Investigation of the canine elbow joint innervation in 100 joints.PLOS ONE

Dear Dr. Jacewicz,

Thank you for submitting your manuscript to PLOS ONE. After careful consideration, we feel that it has merit but does not fully meet PLOS ONE’s publication criteria as it currently stands. Therefore, we invite you to submit a revised version of the manuscript that addresses the points raised during the review process.

We look forward to receiving your revised manuscript.

Kind regards,

Juan Fernando Vélez García

Academic Editor

PLOS ONE

 [Warsaw University of Life Science]. 

Additional Editor Comments:

Dear authors, based on the reviewers, the manuscript should be improved with minor revisions. Please review each comment of the reviewers and answer them. The manuscript should be improved with more references in the introduction and discussion sections. Use the last edition of the Nomina Anatomica Veterinaria. The figures to support your results should be better. Please improve your photographs and schemes so that your article can be published well to our readers.

Reviewers' comments:

Reviewer's Responses to Questions

**Comments to the Author**

1. Is the manuscript technically sound, and do the data support the conclusions?

Reviewer #1: Yes

Reviewer #2: Yes

2. Has the statistical analysis been performed appropriately and rigorously? 

Reviewer #1: Yes

Reviewer #2: Yes

3. Have the authors made all data underlying the findings in their manuscript fully available?

Reviewer #1: Yes

Reviewer #2: Yes

4. Is the manuscript presented in an intelligible fashion and written in standard English?

Reviewer #1: Yes

Reviewer #2: Yes

5. Review Comments to the Author

Reviewer #1: In the manuscript entitled “Investigation of the canine elbow joint innervation in 100 joints” the Authors describe the course of musculocutaneous, median, radial and ulnar nerves which innervate the elbow joint. The data is gathered from appropriately large group of 50 dogs. The Authors describe the location of the nerve branches, which may contribute to improve surgical procedures on elbow joint in the future.

However, as the reviewer I have a several comments:

- The introduction is quite modest in the description of the studied nerves. In my opinion the description of the nerves should be developed with an indication of their sensory and motor functions, which would shed a light on the significance of these nerves and the study itself.

- In line 86, 151, 278… “The distribution of dogs with a different number…” may be understood that the dogs which have a different number of branches per joint are distributed differently than dogs without different number of branches… The Authors should consider changing the sentence. For example… The distribution/number of dogs in relation to the number of branches per joint…

- It should be pointed if the Figures are of own authorship, otherwise the source should be added

- In Fig.3. there should be a legend or description that a red line demonstrates the nerve

- In Figs 6, 9 and 11. same as above… the nerve branches should be marked

- Line 104… the abbreviation CI is not explained as confidence interval

- The statistical results of chi2 test should be presented in the table 3

- In lines 168 or 203 etc. I wound consider changing phrases like “results reported in [2]” into “results reported in Yilmaz and Bahadir’s study”

- In table 6, in one of the table column, there is untranslated fragment of the text

- In line 288, “respectively” is not necessary

- The References are not prepared in accordance to the PLOS ONE Submission Guidelines

Taking into account the above comments, in my opinion the article entitled: “Investigation of the canine elbow joint innervation in 100 joints” is suitable for publishing in PLOS ONE after minor revision.

Reviewer #2: Dear Authors

The manuscript submitted for review, entitled. "Investigation of the canine elbow joint innervation in 100 joints" is written in understandable (not only for anatomists) language, and the issues raised are sure to interest not only specialists in the field. I have a few minor comments of an editorial nature. Please remove the period at the end of the title. Please remove the abbreviation "lat" throughout the manuscript (e.g. extensor carpi radialis muscle (lat: m. extensor carpi radialis) next to the names of anatomical structures-it is unnecessary. Please, next to the name in English write the name in Latin, the reader will know that it is Latin because it is given in italics. Table 6, column 3 - please translate Polish into English. I think that a weakness of this experience is a very modest list of publications to which the authors referred in preparing this manuscript. I think that after submitting a revised version of the manuscript and supplementing the literature section with new items, the work will be even more interesting.

Regards

6. PLOS authors have the option to publish the peer review history of their article (what does this mean?). If published, this will include your full peer review and any attached files.

Reviewer #1: No

Reviewer #2: No

---

## [Author Response · Author response to Decision Letter 0]

27 Sep 2024

We appreciate reviewer’s comments. Thank you for your time and effort.

2.

The authors received no specific funding for this work.

3.

All relevant data are within the manuscript and its Supporting Information files.

Reviewer #1:

- The introduction is quite modest in the description of the studied nerves. In my opinion the description of the nerves should be developed with an indication of their sensory and motor functions, which would shed a light on the significance of these nerves and the study itself.

Corrections have been made to indicate the functions of motor and sensory nerves.

- In line 86, 151, 278… “The distribution of dogs with a different number…” may be understood that the dogs which have a different number of branches per joint are distributed differently than dogs without different number of branches… The Authors should consider changing the sentence. For example… The distribution/number of dogs in relation to the number of branches per joint.

The sentences have been modified.

- It should be pointed if the Figures are of own authorship, otherwise the source should be added

Photo sources have been added.

- In Fig.3. there should be a legend or description that a red line demonstrates the nerve

- In Figs 6, 9 and 11. same as above… the nerve branches should be marked

A legend has been prepared for the diagrams provided.

- Line 104… the abbreviation CI is not explained as confidence interval

This part was corrected.

- The statistical results of chi2 test should be presented in the table 3

This part was corrected.

- In lines 168 or 203 etc. I wound consider changing phrases like “results reported in [2]” into “results reported in Yilmaz and Bahadir’s study”

These parts have been modified

- In table 6, in one of the table column, there is untranslated fragment of the text

We translated that fragment of the text.

- In line 288, “respectively” is not necessary

The term has been removed.

- The References are not prepared in accordance to the PLOS ONE Submission Guidelines

This part was corrected.

Reviewer #2

Please remove the period at the end of the title.

This part was corrected.

Please remove the abbreviation "lat" throughout the manuscript (e.g. extensor carpi radialis muscle (lat: m. extensor carpi radialis) next to the names of anatomical structures-it is unnecessary. Please, next to the name in English write the name in Latin, the reader will know that it is Latin because it is given in italics.

All manuscript was modified.

Table 6, column 3 - please translate Polish into English.

We translated that fragment of the text.

I think that a weakness of this experience is a very modest list of publications to which the authors referred in preparing this manuscript.

One of the main motives of our research was the significantly small amount of literature data on the innervation of the canine elbow joint. This also affects the bibliography of the work. After receiving the reviews, we added several literature items to make the work more interesting.

---

## [Decision Letter · Decision Letter 1]

16 Oct 2024

PONE-D-24-21054R1Investigation of the canine elbow joint innervation in 100 joints.PLOS ONE

Dear Dr. Jacewicz,

Thank you for submitting your manuscript to PLOS ONE. After careful consideration, we feel that it has merit but does not fully meet PLOS ONE’s publication criteria as it currently stands. Therefore, we invite you to submit a revised version of the manuscript that addresses the points raised during the review process.

**The introduction information should include the description of the branches of each nerve to the elbow joint. Hermanson et al. (2020) describe the innervation to the elbow joint and include excellent figures of these branches. You can include a better introduction and review more recent references in veterinary anatomy and studies about locoregional blocks and surgeries involving the nerves. The redaction in English and Latin should improve. Your redaction does not agree with the Nomina Anatomica Veterinaria, for example, you use terms that are not used in the thoracic limb, such as forelimb, foreleg, forepaw, ventral, behind, ulnar loop, and olecranon process. I recommend reviewing your redaction by an English native colleague. I recommend below more recent references that could be included to improve the introduction and discussion of your article. Among them, you can review these:**

**Konig, H. E., & Liebich, H. G. (2020). Veterinary anatomy of domestic animals. *Textbook and color atlas*.**

**Baljit Singh, B. S. (2017). *Dyce, Sack, and Wensing's textbook of veterinary anatomy* (No. Ed. 5, pp. 872-pp).**

**Orsini, J. A., Grenager, N. S., & De Lahunta, A. (Eds.). (2021). *Comparative veterinary anatomy: a clinical approach*. Academic Press.**

**If you use the Latin name of the NAV between parenthesis you do not need to include at the second time. E.g.: The nerve was dissected from the axillary region and then along the brachialis muscle towards the elbow joint. At the joint level, the nerve reaches the cranial side, continuing along the extensor carpi radialis muscle (m. extensor carpi radialis). The radial nerve transmitted articular branches to the cranial quadrant of the capsule, between the extensor carpi radialis muscle (From here is not necessary to include the Latin name for this muscle) and the common digital extensor (m. extensor digitorum communis).**

**-I recommend changing the color of the nerve from red to yellow since this later is the most used in the anatomy veterinary textbooks (Figures 3, 6, 9, 11). The red is used to arteries. The photographs of the figures are blurred (Figures 2, 5, 7, 8, 10).**

We look forward to receiving your revised manuscript.

Kind regards,

Juan Fernando Vélez García

Academic Editor

PLOS ONE

**Additional Editor Comments:**

Dear authors,

The introduction information should include the description of the branches of each nerve to the elbow joint. Hermanson et al. (2020) describe the innervation to the elbow joint and include excellent figures of these branches. You can include a better introduction and review more recent references in veterinary anatomy and studies about locoregional blocks and surgeries involving the nerves. The redaction in English and Latin should improve. Your redaction does not agree with the Nomina Anatomica Veterinaria, for example, you use terms that are not used in the thoracic limb, such as forelimb, foreleg, forepaw, ventral, behind, ulnar loop, and olecranon process. I recommend reviewing your redaction by an English native colleague. I recommend below more recent references that could be included to improve the introduction and discussion of your article. Among them, you can review these:

1. Konig, H. E., & Liebich, H. G. (2020). Veterinary anatomy of domestic animals. Textbook and color atlas.

2. Baljit Singh, B. S. (2017). Dyce, Sack, and Wensing's textbook of veterinary anatomy (No. Ed. 5, pp. 872-pp).

3. Orsini, J. A., Grenager, N. S., & De Lahunta, A. (Eds.). (2021). Comparative veterinary anatomy: a clinical approach. Academic Press.

If you use the Latin name of the NAV between parenthesis you do not need to include at the second time. E.g.: The nerve was dissected from the axillary region and then along the brachialis muscle towards the elbow joint. At the joint level, the nerve reaches the cranial side, continuing along the extensor carpi radialis muscle (m. extensor carpi radialis). The radial nerve transmitted articular branches to the cranial quadrant of the capsule, between the extensor carpi radialis muscle (From here is not necessary to include again the Latin name to this muscle) and the common digital extensor (m. extensor digitorum communis).

-I recommend changing the color of the nerve from red to yellow since this later is the most used in the anatomy veterinary textbooks (Figures 3, 6, 9, 11). The red is used to arteries. The photographs of the figures are blurred (Figures 2, 5, 7, 8, 10). The figure 2 is unnecessary, it only shows the ulnar nerve and your study is about the branches.

Reviewers' comments:

Reviewer's Responses to Questions

**Comments to the Author**

1. If the authors have adequately addressed your comments raised in a previous round of review and you feel that this manuscript is now acceptable for publication, you may indicate that here to bypass the “Comments to the Author” section, enter your conflict of interest statement in the “Confidential to Editor” section, and submit your "Accept" recommendation.

Reviewer #1: All comments have been addressed

Reviewer #2: All comments have been addressed

2. Is the manuscript technically sound, and do the data support the conclusions?

Reviewer #1: Yes

Reviewer #2: Yes

3. Has the statistical analysis been performed appropriately and rigorously? 

Reviewer #1: Yes

Reviewer #2: Yes

4. Have the authors made all data underlying the findings in their manuscript fully available?

Reviewer #1: Yes

Reviewer #2: Yes

5. Is the manuscript presented in an intelligible fashion and written in standard English?

Reviewer #1: Yes

Reviewer #2: Yes

6. Review Comments to the Author

**Reviewer #1:** The manuscript entitled “Investigation of the canine elbow joint innervation in 100 joints” has been improved by the Authors. The Authors have highlighted the significance and the course of musculocutaneous, median, radial and ulnar nerves which innervate the elbow joint. My comments were taken into consideration and the text was corrected in a way that the article represents a greater scientific value. However there are still some minor corrections to be done:

- Line 52, there is double parenthesis

- Some of Latin names are in italics and some not. It should be unified.

- Line 206, double dot

- Line 227, the sentence should be changed. There is: “In results reported… it was reported.

In my opinion the article entitled: “Investigation of the canine elbow joint innervation in 100 joints” is suitable for publishing in PLOS ONE after minor revision.

**Reviewer #2:** (No Response)

7. PLOS authors have the option to publish the peer review history of their article (what does this mean?). If published, this will include your full peer review and any attached files.

Reviewer #1: No

Reviewer #2: No

---

## [Author Response · Author response to Decision Letter 1]

8 Dec 2024

We appreciate reviewer’s comments. Thank you for your time and effort.

The introduction information should include the description of the branches of each nerve to the elbow joint.

The course of the nerves to the elbow joint was described according to the guidelines.

The introduction has been modified and the indicated titles of literature and articles on regional anesthesia of the nerves innervating the elbow joint have been included.

The redaction in English and Latin should improve.

This article has been edited in accordance with the Nomina Anatomica Veterinaria.

Suggestions for Latin names have been included in the article.

I recommend changing the color of the nerve from red to yellow since this later is the most used in the anatomy veterinary textbooks (Figures 3, 6, 9, 11)

The colors of the marked nerves have been changed as suggested.

The photographs of the figures are blurred (Figures 2, 5, 7, 8, 10).

Efforts have been made to improve the quality of the above photos, unfortunately, these are the only photos we have.

Reviewer #1

Line 52, there is double parenthesis

The change was considered.

Some of Latin names are in italics and some not. It should be unified.

The change was considered.

Line 206, double dot

The change was considered.

Line 227, the sentence should be changed. There is: “In results reported… it was reported.

The change was considered.

---

## [Editor Report · Decision Letter 2]

11 Dec 2024

Investigation of the canine elbow joint innervation in 100 joints.

PONE-D-24-21054R2

Dear Dr. Jacewitz,

We’re pleased to inform you that your manuscript has been judged scientifically suitable for publication and will be formally accepted for publication once it meets all outstanding technical requirements.

Kind regards,

Juan Fernando Vélez García

Academic Editor

PLOS ONE

Additional Editor Comments:

Dear authors, the manuscript was improved based on my last suggestions and comments. It is ready to publish. Congratulations.
---

## [Editor Report · Acceptance letter]

16 Jan 2025

PONE-D-24-21054R2 

PLOS ONE

Dear Dr. Jacewicz, 

I'm pleased to inform you that your manuscript has been deemed suitable for publication in PLOS ONE. Congratulations! Your manuscript is now being handed over to our production team.

Kind regards, 

on behalf of

Professor Juan Fernando Vélez García 

Academic Editor

PLOS ONE